# Application of Artificial Intelligence in Lung Cancer

**DOI:** 10.3390/cancers14061370

**Published:** 2022-03-08

**Authors:** Hwa-Yen Chiu, Heng-Sheng Chao, Yuh-Min Chen

**Affiliations:** 1Department of Chest Medicine, Taipei Veterans General Hospital, Taipei 112, Taiwan; chiuhwayen@gmail.com (H.-Y.C.); ymchen@vghtpe.gov.tw (Y.-M.C.); 2Institute of Biophotonics, National Yang Ming Chiao Tung University, Taipei 112, Taiwan; 3Division of Internal Medicine, Hsinchu Branch, Taipei Veterans General Hospital, Hsinchu 310, Taiwan; 4School of Medicine, National Yang Ming Chiao Tung University, Taipei 112, Taiwan; 5Institute of Biomedical Informatics, National Yang Ming Chiao Tung University, Taipei 112, Taiwan

**Keywords:** artificial intelligence, machine learning, lung cancer, radiomics, whole slide imaging, survival prediction

## Abstract

**Simple Summary:**

Lung cancer is the leading cause of malignancy-related mortality worldwide. AI has the potential to help to treat lung cancer from detection, diagnosis and decision making to prognosis prediction. AI could reduce the labor work of LDCT, CXR, and pathology slides reading. AI as a second reader in LDCT and CXR reading reduces the effort of radiologists and increases the accuracy of nodule detection. Introducing AI to WSI in digital pathology increases the Kappa value of the pathologist and help to predict molecular phenotypes with radiomics and H&E staining. By extracting radiomics from image data and WSI from the histopathology field, clinicians could use AI to predict tumor properties such as gene mutation and PD-L1 expression. Furthermore, AI could help clinicians in decision-making by predicting treatment response, side effects, and prognosis prediction in medical treatment, surgery, and radiotherapy. Integrating AI in the future clinical workflow would be promising.

**Abstract:**

Lung cancer is the leading cause of malignancy-related mortality worldwide due to its heterogeneous features and diagnosis at a late stage. Artificial intelligence (AI) is good at handling a large volume of computational and repeated labor work and is suitable for assisting doctors in analyzing image-dominant diseases like lung cancer. Scientists have shown long-standing efforts to apply AI in lung cancer screening via CXR and chest CT since the 1960s. Several grand challenges were held to find the best AI model. Currently, the FDA have approved several AI programs in CXR and chest CT reading, which enables AI systems to take part in lung cancer detection. Following the success of AI application in the radiology field, AI was applied to digitalized whole slide imaging (WSI) annotation. Integrating with more information, like demographics and clinical data, the AI systems could play a role in decision-making by classifying EGFR mutations and PD-L1 expression. AI systems also help clinicians to estimate the patient’s prognosis by predicting drug response, the tumor recurrence rate after surgery, radiotherapy response, and side effects. Though there are still some obstacles, deploying AI systems in the clinical workflow is vital for the foreseeable future.

## 1. Introduction

Lung cancer constitutes the largest portion of malignancy-related deaths worldwide [1]. It is also the leading cause of malignancy-related death in Taiwan [2,3]. The majority of the patients diagnosed with lung cancer are in the late-stage, and therefore have a poor prognosis. In addition to the late stage at diagnosis, the heterogeneity of imaging features and histopathology of lung cancer also makes it a challenge for clinicians to choose the best treatment option.

The imaging features of lung cancer vary from a single tiny nodule to ground-glass opacity, multiple nodules, pleural effusion, lung collapse, and multiple opacities [4]; simple and small lesions are extremely difficult to detect [5]. Histopathological features include adenocarcinoma, squamous cell carcinoma, small cell carcinoma, and many other rare histological types. The histology subtypes vary even more. For example, at least six common subtypes and a total of eleven subtypes of adenocarcinoma were reported in the 2015 World Health Organization classification of lung tumors [6], with more subtypes added to the 2021 version [7]. Treatment options are heavily dependent on the clinical staging, histopathology, and genomic features of the lung cancer. In the era of precision medicine, clinicians need to collect all the features and make a decision to administer chemotherapy, targeted therapy, immunotherapy, and/or combined with surgery or radiotherapy.

Whether to treat or not to treat the disease is always a question in daily practice. Clinicians would like to know the true relationship between the observations and interventions (inputs) and the results (outputs). In other words, to find a model for disease detection, classification, or prediction. Currently, this knowledge is based on clinical trials and the experience of doctors. This exhausts the doctors in reading images and/or pathology slides repeatedly to make an accurate diagnosis. Reviewing charts to determine the best treatment options for patients also consumes a considerable amount of time. A good prediction/classification model would simplify the entire process. Here, artificial intelligence(AI) is introduced.

AI is a general term that does not have a strict definition. AI is an algorithm driven by existing data to predict or classify objects [8]. The main components include the dataset used for training, pretreatment method, an algorithm used to generate the prediction model, and the pre-trained model to accelerate the speed of building models and inherit previous experience. Machine learning (ML) is a subclass of AI, and is the science of obtaining algorithms to solve problems without being explicitly programmed, including decision trees (DTs), support vector machines (SVMs), and Bayesian networks (BNs). Deep learning is a further subclass of ML, featured with multiple layered ML, achieving feature selection and model fitting at the same time [9]. The hierarchical relationship between those definitions is displayed in Figure 1.

However, to develop such a model, a large amount of computation is required. In the past, building a multidimensional algorithm for image analysis has taken hours, or even days, for the human brain. The large computational power required becomes a significant obstacle in creating a sophisticated prediction model. The booming computational power of chip technology and software optimization makes large and sophisticated calculations easier to achieve [10,11]. When a large matrix can be computed in a short time, it is possible to develop models that are much more complex than linear regression or logistic regression. DTs and SVMs were used to build models in the ML era around the year 2000 [12]. By estimating the probability, BNs were used to select treatments by predicting survival [13]. In recent years, deep learning models, including artificial neural networks (ANNs), convolutional networks (CNNs), recurrent neural networks (RCNNs), long-term and short-term memory (LSTM) [14], and generative adversarial networks (GANs) [15], have outperformed most old models and are thus widely used in research and commercial fields [16].

In the 21st century, human life has been largely integrated with AI, and this trend also extends to the medical field. The heterogeneity of lung cancer makes it the best field for AI application. A large number of studies have reported the application in lung nodule detection, diagnostic application in histopathology, disease risk stratification, drug development, and even prognosis prediction. In this article, we present a narrative review of AI applications in lung cancer by introducing AI models first and then reported applications according to the clinical workflow: screening, diagnosis, decision making, and prognosis prediction. Table 1 listed the potential AI application fields in lung cancer.

## 2. AI Models

Numerous AI models are constructed with different algorithms are published nowadays. Generally, the AI models can be divided into: supervised learning, unsupervised learning, semi-supervised learning [9], and reinforcement learning (Figure 2).

### 2.1. Supervised Learning

In supervised learning, researchers need to prepare the labeled dataset with both inputs and desired outputs (answers) to train the algorithm. It is suitable to solve prediction problems, such as classification and regression. The architecture of the algorithms varies. Researchers can use multiple binary nodes to create DTs as a classifier, or find a plane in a multidimensional space as a SVM classifier. Bayesian classifiers used input data to calculate the probability of correct classification. With the probability calculated from the above-mentioned algorithm, researchers can turn the answer into a continuous variable to solve regression problems and vice versa. Most AI applications predicting survival [13,59], cancer risk [34,35,36,37,38,39], nodule detection [22,23], and nodule characteristics [33] are based on supervised learning.

### 2.2. Unsupervised Learning

In unsupervised learning, the algorithm divides the samples according to the inputs by itself. Labeled data are not necessary. It is suitable to do clustering, to find associations between samples, and to do dimensionality reduction. For example, cluster analysis was used to find oncogenes in lung cancer [67,68].

### 2.3. Semi-Supervised Learning

Though supervised learning provides a more accurate algorithm, the labeled data are relatively rare, and the labeling process is labor intensive. Unsupervised learning can adopt unlabeled data but the algorithm is less accurate. Therefore, semi-supervised learning could have both of the advantages when using supervised learning to generate a labeling tool and use supervised learning to generate a large scaled labeled dataset for further training [52].

### 2.4. Reinforcement Learning

Reinforcement learning is a reward-based system. The algorithm evolves as it interacts with the environment (dataset). A reward function is used to adjust the algorithm or the network. This type of AI is famous for playing chess, shogi, and Go through self-play [69] or generating data with GANs [70]. With this technique, researchers can develop a self-evolving AI for nodule hunting on CT images and achieve better accuracy [15,71,72].

In conclusion, there is no best method to build AI models for all. The best method should be tailored according to the clinical question and the dataset used for training.

## 3. Screening

Approximately 7% of patients diagnosed with lung cancer are asymptomatic [73], and more than half of the patients who underwent lung cancer resection were asymptomatic [74]. Several attempts at screening have been made, including imaging, sputum cytology [75,76], blood test screening [77,78], and breath test [79,80]. However, only image screening is able to provide the relevant clues. Although chest X-rays (CXRs) are widely used clinically, low-dose computed tomography (LDCT) is the only method that has been proven to diagnose lung cancer earlier and extend the survival of lung cancer patients [81,82].

The reading workflow of repetitive imaging provides room for AI to participate, because human eyes become sore and images start to blur after reading images for a long time. Furthermore, mistakes in reading CXR or LDCT images occur, and it constitutes a large number of malpractice law suits [83]. Though experts were shown to detect more pulmonary nodules on CXRs [84], approximately 20% of lung nodules <3 cm are missed by radiologists [85]. In the 21st century, the prediction accuracy of pulmonary nodules on CXRs has improved with the computer-aided diagnosis systems or AI-based programs. The sensitivity of radiologists improves from 65.1% to 70.3% with the assistance of AI and the false negative rate decreases from 0.2 to 0.18, changing the diagnosis in 6.7% of the cases [17]. In CT images, the sensitivity of lung nodules were more than 90% spotted by AI-based programs [23]. Integrating AI into lung cancer screening protocol is an ongoing event.

### 3.1. DICOM Format

Digital imaging and communications in medicine (DICOM) is the standard format for image restoration and transfer to enable communication between different servers, manufacturers, and hospitals [86]. The DICOM not only carries pixel data of the image file but also a patient identification number, image type, machine-related parameters, and other information in a format managed by the Medical Imaging and Technology Alliance, a division of the National Electrical Manufacturers Association. After its first publication in 1993, DICOM changed the workflow of radiology, allowing image data to be transmitted quickly and to be analyzed by computers. Later, huge datasets were established for data sharing, model training, or as a benchmark for model testing, and are shown in Table 2.

### 3.2. CXR

CXRs are the most frequently used imaging modality in the medical field. With 0.1 mSv radiation exposure, similar to 10 days of natural background radiation, CXR provides a good examination of the patient’s thorax. Far before digital imaging, the computer-aided diagnosis(CAD) system for CXR has been developed since the 1960s [97]. Image features, such as shape, size, intensity, and texture, must be manually labeled before being sent for further analysis. In the digital era, computers can directly analyze images. By computing the image pixel-by-pixel, radiomics expands the definition of image features from a computer perspective. By computing the image texture and density using different mathematical techniques, the region of interest area can be converted to higher dimension data and expressed as a huge matrix. Because the principle of radiomics is math, the various image qualities of CXR give the computer another task. To obtain accurate radiomics data, image augmentation is an important procedure before nodule detection [98], including pre-processing, lung segmentation [88], and rib suppression [94].

Further malignancy/benign classification was performed using a different algorithm. DT-based algorithms were widely used to analyze these features before 2011. Later, deep-learning-based algorithms demonstrated their power in image analysis. CheXNet, a radiologist-level deep learning algorithm trained on Chest-Xray14, one of the largest CXR databases in the world, exceeds radiologist performance in the detection of 14 pulmonary diseases, including lung nodules and lung masses with an area under the receiver operating curve (AUROC, AUC) 0.78 and 0.87, respectively [18]. Further deep learning models pushed the sensitivity to 0.83 at a false-positive rate of 0.2 per CXR [19]. Currently, several software programs have been approved by the FDA [17,20,21].

### 3.3. Chest CT

CT technology provides a noninvasive method to explore the 3-dimensional structure of the thorax. As the technology advanced, the radiation exposure has reduced from 7 mSv (conventional chest CT) to 1.6 mSv (LDCT). Screening with LDCT showed an approximately 20% mortality reduction in two large randomized control trials: the National Lung Screening Trial (NLST) [81] and the Dutch-Belgian Randomized Lung Cancer Screening Trial (Dutch acronym: NELSON study) [82]. The Multicentric Italian Lung Detection (MILD) trial showed that prolonged LDCT screening for more than five years reduced lung cancer mortality and overall mortality at ten years. These trials boosted the demand for image reading. The application of AI in LDCT reading can help radiologists reduce laborious work, minimize reader variability, and improve screening efficiency [99,100]. The main task for AI application in LDCT reading is the same as in CXR: nodule detection and classification/malignancy prediction. However, unlike CXR, the radiodensity of LDCT is based on an international standard scale with a Hounsfield unit (HU) and fixed resolution. The preprocessing of the CT images focused on denoising, resizing, and lung segmentation.

Many studies have used AI algorithms to detect lung nodules in chest CT images [101,102]. Because they used different models on different datasets and evaluated the models with different benchmarks, such as sensitivity, specificity, AUC, and accuracy, it was difficult to evaluate the models scientifically. A series of grand challenges, such as the Automated Nodule Detection 2009 (ANODE09) study [22] and the Lung Nodule Analysis 2016 (LUNA16) challenge [23] were conducted to find the benchmark model of nodule detection on CT. The best algorithm achieved a sensitivity of 97.2% at one false-positive rate per scan. AI has also been proven to help radiologists increase the sensitivity of nodule detection [24,25,26] and reduce interpretation time. AI has proven to be a good concurrent reader or a second reader. It was noticeable that consumer AI was not approved as the first reader, in case that the radiologists may not have the chance to access to the AI-missed nodules.

Lung nodule classification and malignancy prediction are important tasks in nodule detection. Nodules are classified according to their texture as solid, part-solid, or non-solid, and their size. An AI model trained on the MILD trial [103] dataset and externally validated on the Danish Lung Cancer Screening Trial (DLCST) [104] showed that AI performed equivalently to a human expert on differential six textures (sold, part-solid, non-solid, calcified, perifissural, and speculated) [27]. The classification was then used to predict the malignancy probability as recommended by the Lung CT Screening Reporting and Data System (LUNG-RADS) [105] and the Fleischner guideline [106]. Traditionally, researchers have used AI to classify lung nodules as a feature extraction step, and then go for malignancy prediction. Later, researchers substituted the classification step with radiomics’ feature extraction to increase prediction accuracy [28,29,30].

Similar to nodule detection on CT, challenges were conducted to compare the prediction models. In 2015, the LUNGx Challenge for computerized lung nodule classification was established. Ten teams sent their reports with AUC between 0.50 and 0.68, and only three of them performed statistically better than a random guess, while radiologists performed with AUC between 0.70–0.85. The technology advanced rapidly in the ISBI 2018 Lung Nodule Malignancy Prediction Challenge, and 11 participants completed the challenge with an AUC between 0.70–0.91. The top five participants used deep learning models with AUC between 0.87–0.91 without significant differences from each other [31]. The accuracy was 93% with a sensitivity of 82% and precision of 84% based on the weighted voting method of the autoencoder, ResNet, and handicraft features [32], and 96% with deep convolutional network learning (DCN) [33].

### 3.4. Novel Screening Tests

Genomics [107], proteomics, microbiomes [108], and exhaled breath [109] are novel screening tools for lung cancer [110]. Although these screening methods yield a large set of signals for each patient, an advanced algorithm would elevate the diagnostic yield.

Genomics is one of the most popular topics in oncology. With the polymerase chain reaction (PCR) amplification method and related technology, scientists can now analyze the whole genome [111], exome, transcriptome [112], and epigenome of cancers and produce large sets of information about patients and their tumors. By analyzing whole-genome and whole-transcriptome sequencing data from treatment-naïve patients in The Cancer Genome Atlas [113] (TCGA), the machine learning model successfully discriminated cancer-free healthy controls from patients with cancer [34].

Proteins and other metabolites acquired from plasma and urine samples are relatively easy to obtain and have been studied as a screening tool for lung cancer for decades. To handle the extremely large number of variables produced by proteomics, researchers have used machine learning methods to reduce dimensionality and feature selection [35,36]. In 2003, the machine learning methods were applied to analyze 1676 original and 124 prescreened mass spectra data from 24 diseased and 17 healthy specimens, and researchers successfully built predictive models to discriminate lung cancer specimens from healthy specimens. However, the most accurate predictions were obtained using less interpretable models [35]. Following this idea, the urine proteome combined with machine learning analysis successfully established models that can discriminate lung cancer samples not only from healthy ones, but also from samples from other cancers [36].

Exhaled breath is composed of volatile organic compounds (VOCs) and exhaled breath condensates [79,109]. To date, more than 3000 VOCs have been identified to be related to lung cancer [114], however, not a single VOC could be accurate enough for diagnosis. Therefore, a composite prediction model is a way to solve this problem, in addition to escalating sensor technology. In 2018, a logistic regression model that was able to discriminate patients with lung cancer in both smokers (sensitivity, 95.8%; specificity, 92.3%), and non-smokers (sensitivity, 96.2%; specificity, 90.6%) [37]. Using this, an SVM model [38] and an ANN model [39] were created.

## 4. Diagnosis

When a nodule is detected, clinicians must know the properties of the lung nodule. The gold standard is to acquire tissue samples via either biopsy or surgery. The image features provide a way to guess the properties of the lung nodule by radiomics as mentioned in the previous section. Aside from imaging features, the histopathological features also affect further treatment. Following the path of digital radiology, whole slide imaging (WSI) has opened the trend of digital histopathology. With digitalized WSI data, AI can help pathologists with daily tasks and beyond, ranging from tumor cell recognition and segmentation [47], histological subtype classification [48,49,50,51], PD-L1 scoring [52], to tumor-infiltrating lymphocyte (TIL) count [53].

### 4.1. Radiomics

Following the idea of radiomics in nodule detection and malignancy risk stratification, radiomics was applied to predict the histopathological features of lung nodules/masses [40]. Researchers used logistic regression of radiomics and clinical features to distinguish small cell lung cancer from non-small cell lung cancer with an AUC of 0.94 and an accuracy of 86.2% [41]. The LASSO logistic regression model was used to classify adenocarcinomas and squamous cell carcinomas in the NSCLC group [42]. Further molecular features such as Ki-67 [43], epidermal growth factor receptor (EGFR) [44], anaplastic lymphoma kinase (ALK) [45], and programmed cell death 1 ligand, (PD-L1) [46] were also shown to be predictable with AI-analyzed radiomics, a non-invasive and simple method.

### 4.2. WSI

The emergence of WSI is a landmark in modern digital pathology. The WSI depends on a slide scanner that can transform glass slides into digital images with the desired resolution. Once the images are stored on the server, pathologists can view them on their personal computers or handheld devices. Similar to DICOM in diagnostic radiology, in 2017, the FDA approved two vendors for the WSI system for primary diagnosis [115,116]. Meanwhile, the DICOM also planned support for WSI in the PACS systems to facilitate the adaption of digital pathology in hospitals and further information exchange [117,118]. These features enable the building of a digital pathology network to share expertise for consultations and make education across the country possible [119].

Each WSI digital slide is a large image. It may contain more than 4 billion pixels and may exceed 15 GB when scanned with a resolution of 0.25 micrometers/pixel, referred to as 40× magnification [118,120]. With recent advances in AI and DL in image classification, segmentation, and transformation, digitalized WSI provides another broad field to play. There are many applications for deep learning in cytopathology.

### 4.3. Histopathology

Detecting cancerous regions is the most basic and essential task of deep learning in pathology. Some models combine the detection, segmentation, and histological subtyping together [47,48,49]. Accuracy depends on the data quality, quantity, and abundance of the malignant cell differentiation status. It is difficult to perform histological subtyping of lung cancer without special immunohistochemistry (IHC) staining. This causes inter-observer disagreement when reading H&E staining. While the agreement between pathologists came to a Kappa value of 0.485, a trained AI model can achieve a Kappa value of up to 0.525 when compared with a pathologist [48]. In the detection of lymph node metastasis, a well-trained AI model can help reduce human workload and prevent errors [121]. It obviously performs better than a pathologist in a limited time and has a greater detection rate of single-cell metastasis or micro-metastasis [121].

Although WSI with H&E-stained slides is designed to view the morphology of tissues, with the aid of AI, researchers have designed methods to predict specific gene mutations, PD-L1 expression level, treatment response, and even the prognosis of patients. Focusing on lung adenocarcinoma, Coudray et al. developed an AI application using Inception-V3 for the prediction of frequently appearing gene mutations including STK11, EGFR, FAT1, SETBP1, KRAS, and TP53 [50]. The AUC of this prediction reached 0.754 for EGFR and 0.814 for KRAS which can be treated with effective targeted agents. Sha et al., used ResNet-18 as the backbone to predict the PD-L1 status in NSCLC [55]. Their model showed an AUC between 0.67 and 0.81, while different PD-L1 cutoff levels were chosen. They believed that the morphological features may be related to PD-L1 expression level.

Next-generation sequencing (NGS) plays an important role in modern lung cancer treatment [122]. Successful NGS testing depends on a sufficient number of tumor cells and tumor DNA. AI can assist in determining tumor cellularity [123,124]. In addition, a trained AI can help count the immune cells, while the tissue specimen is adequately stained for special surface markers [53]. Since the PD-L1 expression level is the key predictor for immunotherapy in lung cancer, AI has been trained to count the proportion score for PD-L1 expression [52,125]. When properly stained, computer-aided PD-L1 scoring and quantitative tumor microenvironment analysis may meet the requests of pathologists, and eliminate inter-observer variations and achieve precise lung cancer treatment [126].

However, there are several barriers to the translation of AI applications into clinical services. First, AI applications may not work well when applied to other pathology laboratories, scanners, or diverse protocols [127]. Second, most AIs are designed for their own unique functions. Users are requested to launch several applications for different purposes and spend a lot of time transferring the data. Medical devices powered by AI applications require approval by regulations. Most articles and works were in-house studies and laboratory-developed tests. All of these barriers may restrict the deployment of trained AI models in daily clinical practice [119].

### 4.4. Cytology

The WSI for cytology differs from pathology. Cytology slides are not even sliced flat layers. Instead, they have an entire cell on the glass and would be multiple cell layers. Cytologists tend to use the focus function and look into the cells. While digitalizing the cytology glass slide, the focus function was simulated through the Z-stack function and multiple layers of different focus [128,129]. This method yields a larger WSI file, approximately 10 times that of a typical histological case. Multiple image layers also increase complexity and pose challenges to AI applications.

Few articles have discussed cytology, especially those focusing on lung cancer. For thyroid cancer, Lin et al. proposed a DL method for thyroid fine-needle aspiration (FNA) samples and ThinPrep (TP) cytological slides for detecting papillary thyroid carcinoma [130]. The authors did not claim the ability to detect other cell types of thyroid cancer using their method. AI can be performed for various cytology samples from lung cancer patients, including pleural effusion, lymph node aspiration, tissue aspiration samples, and endobronchial ultrasound-guided fine-needle aspiration (EBUS-TBNA) of mediastinal lymph nodes.

## 5. Decision Making and Prognosis Prediction

Oncologists prefer to deploy this technique to its limits. There are many exciting possibilities for the use of the AI technique. By predicting treatment response, including survival and adverse events, AI was proven to have the potential to play a role in clinical decision making [13], to help surgeons choose the specific groups of patients to receive surgery, and to aid radiotherapists in planning the radiation zone.

### 5.1. Medication Selection

In late-stage lung cancer, the identification of driver mutations, PD-L1 expression, and tumor oncogenes affects most the treatment of choice. Using WSI and radiomics, AI could help to identify EGFR mutations [44,50], ALK [45], and PD-L1 expression [46,55,56]. EGFR mutation subtypes have also been classified using radiomic features [57].

Another research point is the use of radiomics, WSI, and clinical data to directly predict cancer treatment response or survival [131]. Dercle et al. retrospectively analyzed the data from prospective clinical trials and found that the AI model based on the random forest algorithm and CT-based radiomic features predicted the treatment sensitivity of nivolumab with an AUC of 0.77, docetaxel with an AUC of 0.67, and gefitinib with an AUC of 0.82 [58]. CT-based radiomics models have also been reported to predict the overall survival of lung cancer [59,60].

One patent application publication declared that using radiomics features of segmented cell nuclei of lung cancer can predict responses to immunotherapy with an AUC up to 0.65 in the validation dataset [132]. Although there is no specific survival prediction model for lung cancer, Ellery et al. developed a risk prediction model using the TCGA Pan-Cancer WSI database including lung cancer [133]. However, the DL algorithm did not provide acceptable prediction power for lung adenocarcinoma or lung squamous cell carcinoma.

### 5.2. Surgery

The gold standard for the treatment of early-stage lung cancer is surgical resection. AI was applied to pre-surgical evaluation [61,62], and prognosis prediction after surgery, and could help identify patients who are suitable to receive adjuvant chemotherapy after surgery [54].

In pre-surgical evaluation, radiologist-level AI could help predict visceral pleural invasion [62], and identify early stage lung adenocarcinomas suitable for sub-lobar resection [61]. After surgery, AI could play a role in predicting prognosis. The model based on radiomic feature nomograms could identify high-risk groups whose postsurgical tumor recurrence risk is 16-fold higher than that of low-risk group [134]. The CNN model pre-trained with the radiotherapy dataset successfully predict a 2-year overall survival after surgery [135]. The model integrating genomic and clinicopathological features was able to identify patients at risk for recurrence and who were suitable to receive adjuvant therapy [54].

### 5.3. Radiotherapy

SBRT is currently the standard of care to treat early-stage lung cancer and/or provide local control for patients who are medically inoperable or refuse surgery. Radiomics-based models have been reported to successfully predict 1-year tumor recurrence via CT scans performed after 3 and 6 months of SBRT [63]. Lewis and Kemp also developed a model trained on TCGA dataset to predict cancer resistance to radiation [64]. As a well-known side effect of radiotherapy, radiation pneumonitis can be lethal, and clinicians would like to prevent this situation. The AI model based on pretreatment CT radiomics was superior to the traditional model using dosimetric and clinical predictors in predicting radiation pneumonitis [65]. Another ANN algorithm trained with radiomics extracted from a 3D dose map of radiotherapy has been shown to predict the acute and late pulmonary toxicities with an accuracy of 0.69 [66]. A well-designed prediction model for radiation pneumonitis may help to prevent radiation pneumonitis in the future.

## 6. Future Development

The future of AI applications in lung cancer could focus on integration and applications. First, because AI is a data-driven technology, scientist can integrate small datasets to create large data sets for training. However, regulations regarding data sharing are a huge obstacle for researchers. Federated learning, a method that shares the trained parameters rather than sharing the data, is a simple solution [136,137]. In federated learning, the models were trained at each different hospitals separately and only the trained models were sent to the main server, so that the main server does not touch the raw data directly. The final model was then reported back to individual hospitals (Figure 3).

Second, most previous researches were conducted by separate specialists and focused on separated fields such as radiology, pathology, surgery, or clinical oncology. However, integrating all aspects such as radiology, pathology, demographics and clinical data, and both old and new technologies could better reflect reality. The combination of different features also helps researchers build predictive models [138,139]. This brings about the idea of multi-omics [140,141] or “Medomics” [40]. Similar to multidisciplinary teams in clinical lung cancer treatment [142,143], the combination of different domain knowledge and multidisciplinary integration is worth pursuing in the future.

Apart from improvement in model accuracy by increasing the training sample size and multidisciplinary integration, another issue is the application of AI programs. Although the studies above all showed the promising results of applying AI in lung cancer and some products were approved by the FDA [17,20,21,116,144], real implementation of the clinical workflow is rare. The user interface, speed of data analysis, expanse of the AI program, internet bandwidth, and resources consumed by the AI program are all barriers to real-world applications. More infrastructure needs to be constructed before we can enter the AI-assisted world.

## Figures and Tables

**Figure 1 cancers-14-01370-f001:**
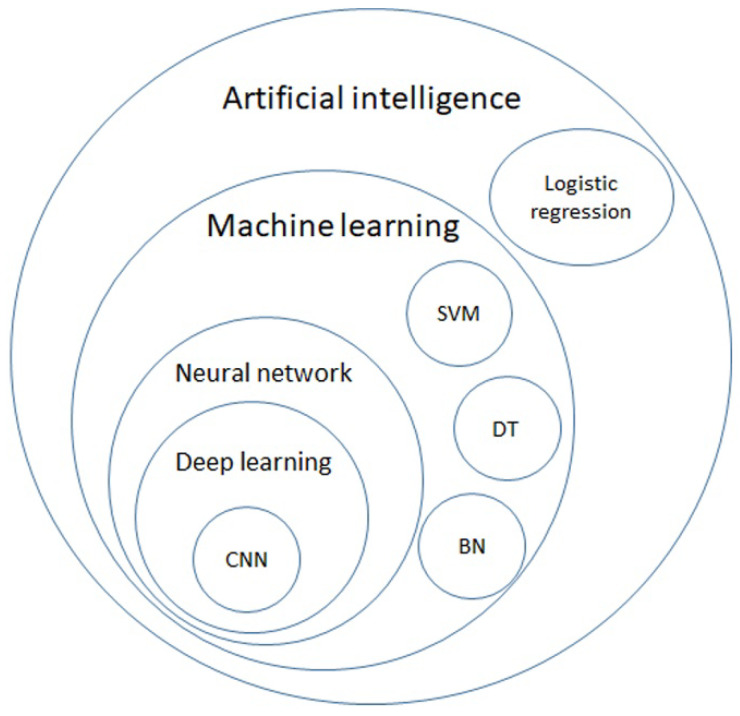
Venn diagram of artificial intelligence (AI), machine learning (ML), neural network, deep learning, and further algorithms in each category. AI is a general term for a program that predicts an answer to a certain problem, where one of the conventional methods is logistic regression. ML learns the algorithm through input data without explicit programming. ML includes algorithms such as decision trees (DTs), support vector machines (SVMs), and Bayesian networks (BNs). By using each ML algorithm as a neuron with multiple inputs and a single output, a neural network is a structure that mimics the human brain. Deep learning is formed with multiple layers of neural networks, and convolutional neural network (CNN) is one of the elements of the famous architecture.

**Figure 2 cancers-14-01370-f002:**
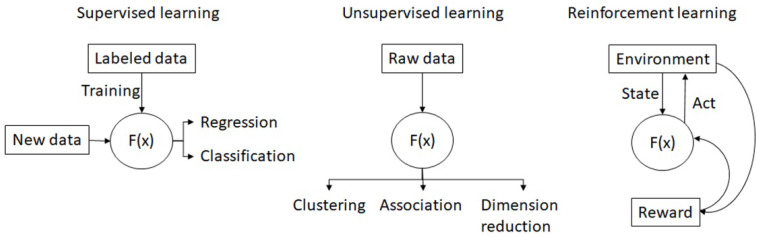
The concept map of supervised learning, unsupervised learning and reinforcement learning.

**Figure 3 cancers-14-01370-f003:**
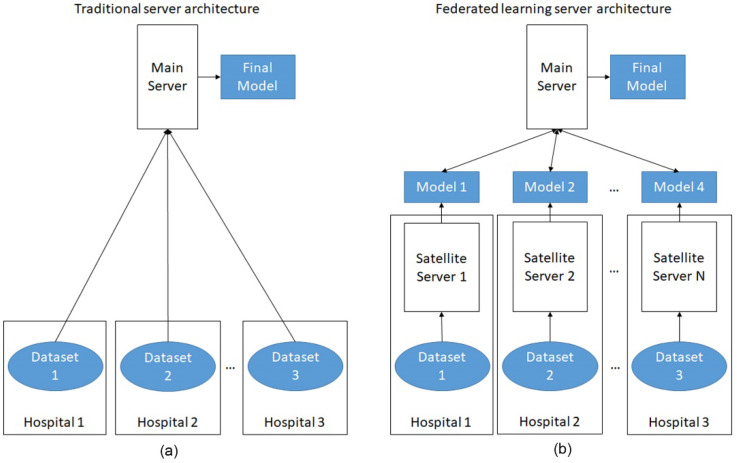
The comparison of traditional AI server architecture and federated learning server architecture. (**a**) In traditional server architecture, the main server processes all the raw data at the same site, leading to concerns about privacy; (**b**) In federated learning, the datasets are processed at each individual site and only the trained models are shared with the main server. Privacy of each dataset is protected.

**Table 1 cancers-14-01370-t001:** Summary of AI application fields.

Screening	Diagnosis	Treatment
Radiology: CXR [17,18,19,20,21] CXR [17,18,19,20,21] LDCT [22,23,24,25,26,27,28,29,30,31,32,33] Novel tools: Genomics [34] Genomics [34] Proteomics [35,36] Exhaled breath [37,38,39]	Risk prediction: Radiomics [40,41,42,43,44,45,46] WSI [47,48,49,50,51,52,53] WSI [47,48,49,50,51,52,53] Genomics [50,54]	Tumor property classification: Drug selection [44,45,46,50,55,56,57] Prognosis prediction: Drug treatment response [58,59,60] Post-Surgery recurrence [54,61,62] Radiotherapy response [63,64] Side effect estimation: Radiation pneumonitis [65,66]

CXR: Chest X-ray, LDCT: low-dose computed tomography, WSI: whole slide imaging.

**Table 2 cancers-14-01370-t002:** Summary of frequently used datasets for model training.

Database	Year	Material	Volume	Features
JSRT [87]	1998	CXR	154	Contains 100 CXRs with malignant nodule, 54 CXRs with benigh nodule, and 93 normal CXRs
Shenzhen CXR set [88]	2012	CXR	662	Contains 326 normal CXRs, and 336 CXRs with tuberculosis. Ribs were labeled.
Montgomery CXR set [88]	2014	CXR	138	Contains 80 normal CXRs, and 58 CXRs with tuberculosis. Ribs were labeled.
ChestXray8 [89]	1992–2015	CXR	108,948	Classified into 8 features: atelectasis, cardiomegaly, effusion, infiltration, mass, nodule, normal, pneumonia, and pneumothorax
ChestXray14 [89]	1992–2015	CXR		Classified into 14 features: atelectasis, cardiomegaly, consolidation, edema, effusion, emphysema, fibrosis, hernia, infiltration, mass, nodule, pleural thickening, pneumonia, pneumothorax.
PadChest [90]	2009–2017	CXR	>160,000	Labeled with 174 different radiographic findings, 19 differential diagnoses and 104 anatomic locations
LIDC [91]	2011	LDCT	1018	Nodules were annotated and labeled with nodule sizes
LUNA16 [23]	2016	LDCT	888	Adapted from LIDC, with additional nodules found during model training. 1186 lung nodules annotated in 888 CT scans
MIMIC-CXR [92]	2011–2016	CXR	377,110	Classified into 14 labels derived from two natural language processing tools.
ChestXpert [93]	2019	CXR	224,316	Labeled with 14 features: no finding, enlarged cardiom, cardiomegaly, lung opacity, lung lesion, edema, consolidation, pneumonia, atelectasis, pneumothorax, pleural effusion, pleural other, fracture, support devices
VinDr-RibCXR [94]	2020	CXR	18,000	Rib suppression images
RadGraph [95]	2021	CXR	500	Inference dataset of MMIC-CXR and reports
REFLACX [96]	2021	CXR	3032	Labeled by 5 radiologists and synchronized sets of eye-tracking data and timestamped report transcriptions

CXR: chest CX-ray set, JSRT: Japanese Society of Radiological Technology, LIDC: Lung Image Database Consortium, LUNA: LUng Nodule Analysis, REFLACX: Reports and Eye-Tracking Data for Localization of Abnormalities in Chest X-rays.

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
