# Peer review of "Application of Artificial Intelligence in Lung Cancer"

_cancers, 2022, doi:10.3390/cancers14061370_

Round 1

Reviewer 1 Report

They have done all the response based on my concern. I have no further comment.

Reviewer 2 Report

The manuscript is improved in comparison to the previous form, however the spectrum of the review still concerns to many aspects that elaboration is very general. Therefore, it may not meet with the high interest of readers.

This manuscript is a resubmission of an earlier submission. The following is a list of the peer review reports and author responses from that submission.

Round 1

Reviewer 1 Report

The authors introduced the application of AI in lung cancer. It is an interesting topic. And the author clearly described the research achievements of artificial intelligence in the detection, diagnosis, decision making and prognosis prediction of lung cancer. However, I have some minor concerns about the manuscript.

  1. In line 44-45 and line 98-100, please add the lack of corresponding references.
  2. In Screening section, the authors introduced the advantages of artificial intelligence in lung cancer screening. The authors mentioned, “Approximately 20% of lung nodules <3 cm are missed by radiologists”. Whether there have been studied using AI-based models to diagnose pulmonary nodules? I suggest the authors report their prediction accuracy for highlight the advantages of AI.
  3. What is the difference between machine learning and deep learning and AI? Does machine learning and deep learning belong with AI? What are the common deep learning and machine learning algorithms? The authors are suggested to explain it in detail.
  4. In addition, the authors frequently mentioned “AI model”,“machine learning model”. Different studies build different models. The authors should describe in detail the process of building these models and the structure of the models.
  5. The area under the curve often referred to as simply the AUC. Please add full name of AUC.

Author Response

Point 1: In line 44-45 and line 98-100, please add the lack of corresponding references.

Response 1: Corresponding references of image features of lung cancer in line 44-45 were added, and corresponding references of evidence of low dose computer tomography in lung cancer screening(new line 137-140) were added. Thanks for your suggestion.

Point 2: In Screening section, the authors introduced the advantages of artificial intelligence in lung cancer screening. The authors mentioned, “Approximately 20% of lung nodules <3 cm are missed by radiologists”. Whether there have been studied using AI-based models to diagnose pulmonary nodules? I suggest the authors report their prediction accuracy for highlight the advantages of AI.

Response 2: In the study by Quekel et al. in 1999, the radiologists read the CXRs without assistance of AI. We added more lines about the improvement with the AI based models behind and amend the paragraph as below:

“The repetitive imaging reading workflow provides room for the AI to participate, while human eyes become soreness and images start to blur after reading images for a long time. Furthermore, mistakes in reading CXR or LDCT images occur, and it constitutes a large number of malpractice law suits. Though experts were shown to detect more pulmonary nodules on CXRs, approximately 20% of lung nodules <3 cm are missed by radiologists. In the 21st century, the prediction accuracy of pulmonary nodules on CXRs improves with the computer-aided diagnosis systems or AI-based programs. The sensitivity of radiologists improves from 65.1% to 70.3% with the assistance of AI and the false negative rate decreased from 0.2 to 0.18 making the diagnosis changed in 6.7% of the cases. In CT images, the sensitivity of lung nodules were more than 90% by AI-based programs. Integrating AI into lung cancer screening protocol is an ongoing event.”

Point 3: What is the difference between machine learning and deep learning and AI? Does machine learning and deep learning belong with AI? What are the common deep learning and machine learning algorithms? The authors are suggested to explain it in detail.

Response 3: We extended the paragraph and added a Venn diagram to describe the relationship between AI, machine learning and deep learning. Thanks for your suggestion.

Point 4: In addition, the authors frequently mentioned “AI model”,“machine learning model”. Different studies build different models. The authors should describe in detail the process of building these models and the structure of the models.

Response 4: We added a whole section to describe the process of building an AI model, including the material should be prepared, algorithms and further structures of AI models. Thanks for your advices.

Point 5: The area under the curve often referred to as simply the AUC. Please add full name of AUC.

Response 5: The abbreviation was mentioned as AUROC in line 139, we add the abbreviation ‘AUC’ as well as ‘AUROC’ in line 139. Thanks for your suggestion.

Reviewer 2 Report

This review drift, Application of artificial intelligence in lung cancer, by Mr. Hwa-Yen Chiu etc indicates that try to use AI to study the relationship between diagnostics data and therapeutic effect will be beneficial to cancer therapy in the future. Recently, FDA approved several AI programs in CXR and chest CT reading, which enables AI system to take part of lung cancer detection. In addition, genomic background of patients will be related to drug response such as immunotherapy. In the future, AI can be used to predict the drug response to improve the survival rate. So, AI application is highly potential to be used clinically in the future. Several suggestions listed here hope to be advantage to this manuscript.

Comments

  1. Several review about AI and lung cancer have been published recently. For example, Dr. Fukuoka published a review, A narrative review of digital pathology and artificial intelligence: focusing on lung cancer, to introduce how to use AI on lung cancer deeply, including the description, table and figures. In this review manuscript, the authors summary all the studies about AI and diagnosis and therapy of lung cancer, but whole of the introduction just summary them. It will be better to organize all the previous studied by figures or table. I can’t get the integrate knowledge from this review manuscript.
  2. It will be better to focus on one of specific fields such as AI on image or genomic background to do the detail organize whole of the related knowledge, not only summary them.
  3. Overall, the title is too broad to introduce them deeply. In addition, question open needs to be described in detail, which will be beneficial for the study in the future.

Author Response

Point 1: Several review about AI and lung cancer have been published recently. For example, Dr. Fukuoka published a review, A narrative review of digital pathology and artificial intelligence: focusing on lung cancer, to introduce how to use AI on lung cancer deeply, including the description, table and figures. In this review manuscript, the authors summary all the studies about AI and diagnosis and therapy of lung cancer, but whole of the introduction just summary them. It will be better to organize all the previous studied by figures or table. I can’t get the integrate knowledge from this review manuscript.

Response 1: Thanks for your comments and we put the references in table 1 to classify them in order to different applications. Hope this will help to get integrate knowledge about the application fields of AI in lung cancer.

Point 2: It will be better to focus on one of specific fields such as AI on image or genomic background to do the detail organize whole of the related knowledge, not only summary them.

Response 2: This article is intention to bridges the minds between different specialists such as clinicians, radiologists, pathologists, radio-oncologists, technicians, and programmers. We try to report the whole picture of AI application in lung cancer and guide more researchers to find ‘to-do’ details through the references.

Point 3: Overall, the title is too broad to introduce them deeply. In addition, question open needs to be described in detail, which will be beneficial for the study in the future.

Response 3: While we try to bridges different specialists, we have to introduce the whole ecosystem of lung cancer from screening, diagnosis to treatment. Therefore, the title is relatively broad but we think this review article is beneficial for researchers to integrate different knowledge fields. In the last section, we had raised some questions and described some solutions already. We then further added the detail of federated learning and added a metaphor about multi-omics. Thanks for your suggestion.

Reviewer 3 Report

The manuscript proposed by Hwa-Yen Chiu et al. comprehensively summarize the available literature data about AI application in LC. To improve the quality of paper I would like to suggest some minor remarks:

  1. Many paragraphs are left without references – please implement them properly.
  2. Table 1- each potential application of AI should be referred by specific citation.
  3. Please review the pre-prints data base (ex. bioRxiv) where many of new high quality articles related to the field is deposited.

Author Response

Point 1: Many paragraphs are left without references – please implement them properly.

Response 1: We have implanted the references. Thanks for your advice.

Point 2: Table 1- each potential application of AI should be referred by specific citation.

Response 2: We have implanted the references mentioned in the article into Table 1. Thanks for your suggestion.

Point 3: Please review the pre-prints data base (ex. bioRxiv) where many of new high quality articles related to the field is deposited.

Response 3: We have reviewed the pre-prints database and found some interesting articles about digital pathology and multi-omics. They were also put in the references. Thanks for your suggestion.